

# Statistical Evaluation of Well Water Anomalies as Potential Precursors to Large Earthquakes

Yoshiaki Orihara[1]

5  [1] Department of Physics, Tokyo Gakugei University, 4-1-1 Nukuikita-machi, Koganei-shi, Tokyo 184-8501, Japan

*Correspondence to*: Yoshiaki Orihara (orihara@u-gakugei.ac.jp)



**Abstract.** Macroscopic anomalies such as well water fluctuations and unusual animal behaviour have been recorded in old documents and folklore accounts as possible earthquake precursors. However, no reliable earthquake prediction method based on such anomalies has ever been established. One of the reasons is that the same phenomena often occur independent of earthquakes. This study conducted a re-examination using shallow well water observation data recorded for eleven years by a volunteer observation network in Japan. While some critics have claimed that it was impossible to predict an earthquake using well water anomalies, especially shallow ones, the results suggest that some plausible anomalies may serve as true precursors. On the other hand, many anomalies were observed without associated earthquakes, and earthquakes without associated anomalies were also uncovered. The old documents and folklore may contain elements of truth, while unrecorded failures are hidden behind successful accounts. Credible anomalies were limited, and major earthquakes rarely occurred within the same focal region. Furthermore, maintaining systematic observations has been challenging. Even an eleven-year observation period was too short to empirically verify the feasibility of an earthquake prediction method for disaster mitigation. Therefore, continuous support is needed for empirical research on earthquake precursors.

## 1 Introduction

Earthquake (EQ) prediction accurately specifies the time, place, and magnitude ($M$) of an impending EQ. According to Uyeda (2013), "EQ predictions should be short-term predictions based on observable physical phenomena or precursors". Macroscopic anomalies, including unusual well water changes, are candidates for precursors that can be recognized by human senses without the aid of instruments. Numerous groundwater anomalies, such as those associated with water levels, temperatures, and muddy water preceding large EQs, have been reported, especially in China and Japan (e.g., Wakita, 1978; Wu and Zheng, 1984; Roeloffs, 1988; Orihara et al., 2014). Some folklore suggests unusual preseismic well water changes in each place in Japan (Daigo, 1985). Such phenomena are generally thought to be macroscopic anomalies.

In Japan, a volunteer observation network has been systematically monitoring groundwater changes since 1976. This research group is called the "Catfish Club" and is organized by the Hot Springs Research Institute of Kanagawa Prefecture, Japan. They have mainly monitored water levels in shallow wells. Some of them have observed hot spring water temperature anomalies. The main purpose of their observations pertains to short-term EQ prediction. The maximum number of monitoring members in the club was more than two hundred before their activities gradually decreased (Oki and Hiraga, 1988). Currently, there are fourteen members (https://www.onken.odawara.kanagawa.jp/laboratory/catfish-society/).

The Catfish Club has observed anomalous changes prior to EQs such as the 1978 $M$ 7.0 Izu Ohshima Kinkai EQ and the 1978 $M$ 7.4 Miyaji-Oki EQ, and these changes were reported in its newsletters (Suzuki, 1978; Water-level Observation Group of Namazu-no-kai, 1978). These reports mainly pertain to EQs preceded by anomalies. An examination to clarify the correlation between anomalies and EQs should not only consider these cases but should also consider EQs with no preceding anomalies,





anomalies that were not followed by EQs, and cases with no anomalies and no EQs. Fortunately, the newsletters included daily

records called the "Black and White" Score Table for Groundwater Level Observations for eleven years.

Although the criteria for the score tables were unknown, according to the Catfish Club (1977), several experts judged whether each data point contained an anomaly by eliminating changes caused by rainfall, atmospheric pressure, and nearby pumping. Therefore, we considered that the daily records were objective data that were sufficient for re-examination.

The newsletters written in Japanese are owned by the Kanagawa Prefectural Library, and everyone can read them. First, we

transcribed the paper-based data into a digital format for statistical investigation and then verified the folklore accounts that suggested that well water anomalies occurred prior to an impending EQ. Finally, the feasibility of practical EQ prediction was discussed.

## 2 Methods

### 2.1 Monitoring by the Catfish Club

According to Oki and Hiraga (1988), the Catfish Club observed groundwater levels using digital water gauges while simultaneously measuring precipitation and atmospheric pressure. Most of the wells were once used for drinking water and were 5 to 20 metres in depth. They recorded the monitoring time, water levels, weather, air pressure, temperature, rainfall, etc., once or twice a day. The information was recorded on postcards as prescribed and sent to the head office of the Catfish Club. Numerical datasets gathered at the headquarters were graphed. Whether or not the data contained anomalies was judged by

some experts. If the unusual changes remained after eliminating changes caused by rainfall, atmospheric pressure, and nearby pumping of water, they regarded them as true anomalies. The anomalous changes meant deviations from the expected positive or negative trends.

The observation results appeared in the newsletters as tables called the "Black and White" Score Table for Groundwater Level Observations issued every four months. Black circles denoted the days for which anomalies were observed, while white circles

indicated days for which no anomalies were observed. Most of the anomalies presented in this paper indicated unusual groundwater level changes except for two stations where were changes in the groundwater temperature were observed. Some members immediately called the head office when they suspected the observed change was an anomaly. For instance, a few days before the 1978 $M$ 7.4 Miyagi-Oki EQ, a member who lived in Miyagi Prefecture made an emergency call to the headquarters (Suzuki, 1978).

### 2.2 Observation stations and EQ selection

The observation results showed that from February 1, 1977, to March 31, 1987, (3710 days), 234 members conducted the monitoring. We selected the stations used in this study based on the following criteria. First, stations with unknown addresses were excluded. Next, stations with less than two years of observational data were excluded, as at least two years are required





to assess the presence or absence of seasonal variations. Although some stations were close together, their anomalies did not
appear on the same days. We regarded such stations as independent. Consequently, 108 stations remained.

Oki and Hiraga (1988) considered that even EQs with magnitudes below $M$ 5 could be successfully predicted. However, we focused on EQs likely to cause severe damage and thus chose EQs with magnitudes above $M$ 6.0 (depth < 100 km). For the distance to the epicentre, Oki and Hiraga (1988) adopted the following formula to represent the spatial limit between water level precursors ($R$) and the magnitude of an EQ ($M$): $\log R = 0.47 M - 0.73$.

This empirical equality means that the larger the EQ magnitude is, the farther the distance limit of the precursor. The formula also suggests that for $M=6.0$, the distance ($R$) is 123 km, for $M=7.0$ it is 363 km, and for $M=8.0$ it is 1,071 km. On the other hand, Rikitake (1998) claimed that for $M=6.0$, 7.0 and 8.0, $R$ is 39 km, 95 km, and 230 km, respectively, when considering macroscopic anomalous phenomena such as well water anomalies. The detectable distance by Oki and Hiraga (1988) seemed too large. We employed the following relationship between $R$ and $M$: $R = 100$ km for $6.0 \leq M < 6.5$, 150 km for $6.5 \leq M < 7.0$,

200 km for $7.0 \leq M < 7.5$, and 250 km for $7.5 \leq M < 8.0$. By applying the above criteria, the number of selected EQs was reduced 18 (see Table 1). The 18 EQs consisted of 8 with $6.0 \leq M < 6.5$, 4 with $6.5 \leq M < 7.0$, 5 with $7.0 \leq M < 7.5$, and 1 with $7.5 \leq M < 8.0$.

**2.3 Criteria for successful and false alarms**

Although Oki and Hiraga (1988) did not specify a time interval $\Delta T$, i.e., the period between the day when the anomaly was observed and the day when the EQ occurred, we considered $\Delta T$ as the day that the anomaly appeared and the following 7 days; thus, $\Delta T$ represents "alarm days." When anomalies were observed consecutively, all anomaly days and the continuous 7 days were considered alarm days. If another anomaly appeared before the end of the 7-day interval, the alarm days continued (Fig. 1a). When the selected EQ occurred during the alarm day interval, we considered the interval to be "successful alarm days",

whereas the absence of an EQ during this interval resulted in "false alarm days." When a new anomaly appeared on the day following the end of the alarm days, the former and latter were treated as separate intervals (Fig. 1b). For instance, if an EQ occurred during the latter 7-day interval, the former interval was categorized as "false alarm days". In addition, the anomalies observed three days after the EQ were considered as neither successful nor false alarm days because those changes might have occurred due to coseismic effects, and anomalies appearing on the 4th day after the EQ were classified as another anomaly.

Even if there were no observation data only on the day of the EQ or on the day before the EQ, we considered such stations to be able to detect the EQ. On the other hand, if data were missing for both the EQ occurrence day and the day before the EQ, the EQ was excluded from the predictable EQ.



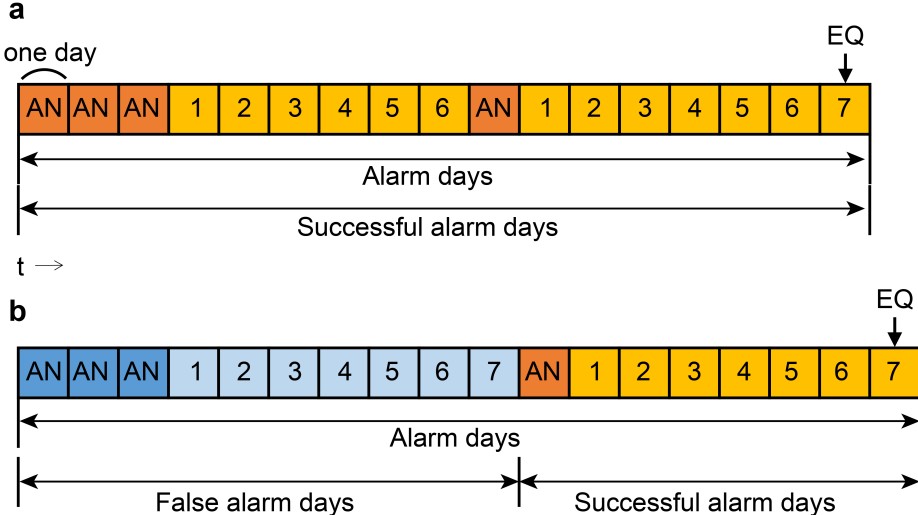

**Fig. 1. Diagram showing the alarm function. Each small square indicates one day. AN indicates the day that an anomaly occurred.**
**In each scenario, an earthquake (EQ) occurred on the final day (indicated on the right). (a) All alarm days were considered successful alarm days because the 4th AN occurred before the end of the former sequence of alarm days. (b) The first ten days in the sequence are considered false alarm days, whereas the latter 8 days are considered successful alarm days.**

## 3 Results and Discussion

In this paper, the observation station name comprises three letters and two numbers. Table 1 shows the number of stations
located within detectable distances from the 18 EQs, the number of affected stations, and the hit rate of each of the 18 EQs.
Note that in this instance "hit" refers to the case when the anomaly appears before a predictable EQ. Ten of the 18 EQs had
prior anomalies, and the other 8 had no prior anomalies. The total numbers of detectable stations and affected stations were
412 and 64, respectively; therefore, the average hit rate was 16%. Fig. 2 shows the locations of the 108 stations and the 18 EQs
with detectable distance circles. Two stations marked in red, KNG80 and SZK05, observed water temperature, and the other
106 stations conducted water level observations. Various information from all 108 stations is included in Table 2. Seventy-
two stations recorded anomaly days, and the other 36 did not. The numbers of stations with 0, 1, 2, 3, 4, 5, 6, and 7 predictable
EQs were 2, 10, 22, 17, 17, 12, 17, and 11, respectively. Two stations, SZK14 and ISK01, did not record both anomaly days
and predictable EQs. Forty-six of the 72 stations had successful alarm days. The anomalies did not exhibit like seasonal
variations at any of the 46 stations. The average of $v$ (the fraction of the prediction failure) (Molchan, 1991; Molchan and
Kagan1992) was 0.83, and the ratio of all successful alarm days to all alarm days was 0.14 (14%).

**Table 1. The 18 selected EQs, the number of stations located within a detectable distance from each EQ, the number of stations with successful hits, and their hit rates.**





| No. | Date | Longitude | Latitude | Depth [km] | Magnitude | Epicentre region | Detectable stations | Hit stations | Hit rate (%) |
|---|---|---|---|---|---|---|---|---|---|
| 1 | 14 Jan. 1978 | 139.250 | 34.767 | 0 | 7 | Izu Ohshima | 65 | 26 | 40 |
| 2 | 20 Feb. 1978 | 142.200 | 38.750 | 50 | 6.7 | Off Miyagi Prefecture | 2 | 2 | 100 |
| 3 | 12 June 1978 | 142.167 | 38.150 | 40 | 7.4 | Off Miyagi Prefecture | 2 | 2 | 100 |
| 4 | 13 July 1979 | 132.050 | 33.850 | 70 | 6 | Coast of Yamaguchi Prefecture | 1 | 1 | 100 |
| 5 | 29 June 1980 | 139.233 | 34.917 | 10 | 6.7 | Izu Peninsula | 87 | 18 | 21 |
| 6 | 25 Sep. 1980 | 140.217 | 35.517 | 80 | 6 | Chiba Prefecture | 58 | 3 | 5 |
| 7 | 19 Jan. 1981 | 142.967 | 38.600 | 0 | 7 | Off Miyagi Prefecture | 2 | 0 | 0 |
| 8 | 23 July 1982 | 141.950 | 36.183 | 30 | 7 | Off Ibaraki Prefecture | 3 | 0 | 0 |
| 9 | 27 Feb. 1983 | 140.152 | 35.940 | 72 | 6 | Ibaraki Prefecture | 35 | 6 | 17 |
| 10 | 26 May 1983 | 139.073 | 40.360 | 14 | 7.7 | Off Akita Prefecture (Sea of Japan) | 2 | 1 | 50 |
| 11 | 8 Aug. 1983 | 139.022 | 35.522 | 22 | 6 | Border of Kanagawa and Yamanashi Prefectures | 67 | 3 | 4 |
| 12 | 7 Aug. 1984 | 132.153 | 32.383 | 33 | 7.1 | Hyuganada | 1 | 0 | 0 |
| 13 | 14 Sep. 1984 | 137.557 | 35.825 | 2 | 6.8 | Nagano Prefecture | 9 | 2 | 22 |
| 14 | 4 Oct. 1985 | 140.155 | 35.872 | 78.4 | 6 | Ibaraki Prefecture | 40 | 0 | 0 |
| 15 | 24 June 1986 | 140.717 | 34.827 | 73.3 | 6.4 | Off Boso Peninsula | 1 | 0 | 0 |
| 16 | 22 Nov. 1986 | 139.522 | 34.550 | 15.1 | 6 | Izu Ohshima | 35 | 0 | 0 |
| 17 | 1 Dec. 1986 | 142.137 | 38.875 | 50.8 | 6 | Off Iwate Prefecture | 1 | 0 | 0 |
| 18 | 9 Jan. 1987 | 141.777 | 39.837 | 71.6 | 6.6 | Coast of Iwate Prefecture | 1 | 0 | 0 |

Notes:

Information on the 18 EQs came from the Japan Meteorological Agency earthquake catalogue. The numbers 1 to 18 correspond to the same numbers in Fig. 2.





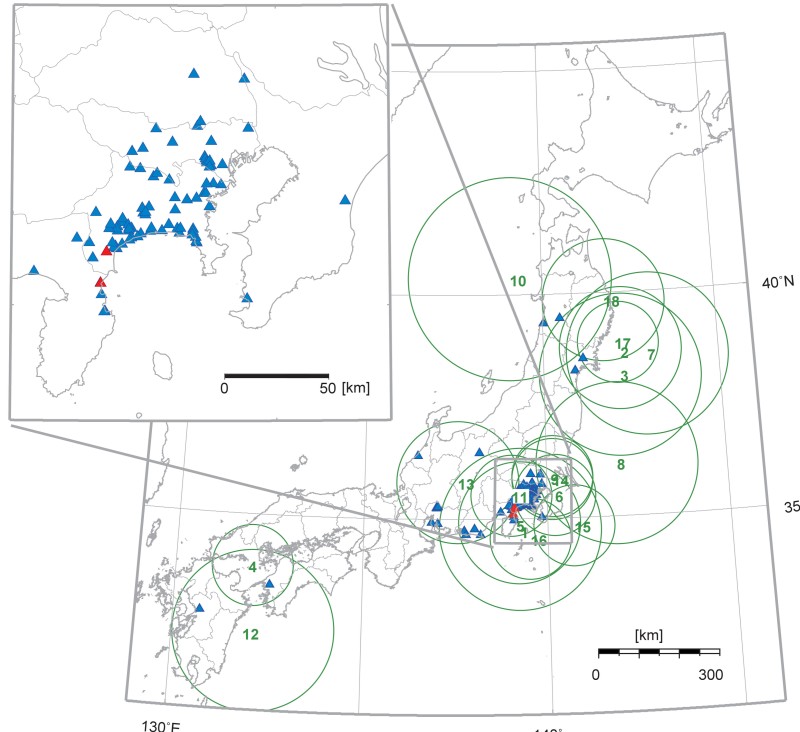

**Fig. 2. Locations of the 108 stations and the 18 EQs considered in this study. The blue triangles denote the 106 stations where the water level was recorded, and the 2 red triangles indicate the stations where the water temperature was monitored. Each green circle is centred at the epicentre of one of the 18 earthquakes and represents the area within which precursory signals could potentially be detected. The numbers 1 to 18 correspond to the EQ numbers in Table 1. The circles are drawn according to the following relationship between $R$ and $M$: $R$ = 100 km for $6.0 \leq M < 6.5$, 150 km for $6.5 \leq M < 7.0$, 200 km for $7.0 \leq M < 7.5$, and 250 km for $7.5 \leq M < 8.0$.**

**Table 2. Information about the observation stations: latitude and longitude, number of observation days, number of anomaly days, number of alarm days, and so on.**

| Station | Latitude (degree N) | Longitude (degree E) | Observed Days | Anomaly Days | Alarm Days | Successful Alarm Days | $\tau : P(A)$ | Predictable EQs [Number] | EQs with Preceding Anomalies [Number] | $\nu : 1 - P(A\|E)$ | Probability Gain ($PG$) | Successful Alarm Rate (%) | $p\text{-value}$ (more than $k$) |
|---|---|---|---|---|---|---|---|---|---|---|---|---|---|
| AKT01 | 39.3517 | 140.0202 | 3701 | 6 | 13 | 4 | 0.0035 | 1 | 1 | 0.00 | 285 | 31 | 0.0035 |
| AKT02 | 39.4623 | 140.4943 | 3431 | 0 | 0 | 0 | | 2 | 0 | 1.00 | | | |



| | | | | | | | | | | | | |
|---|---|---|---|---|---|---|---|---|---|---|---|---|
| MYG01 | 38.5468 | 141.1261 | 3066 | 6 | 20 | 0 | 0.0065 | 2 | 0 | 1.00 | 0 | 0 |
| MYG03 | 38.2837 | 140.8924 | 2127 | 51 | 72 | 35 | 0.0339 | 2 | 2 | 0.00 | 30 | 49 | 0.0011 |
| MYG04 | 38.2836 | 140.8927 | 2287 | 15 | 29 | 6 | 0.0127 | 3 | 2 | 0.33 | 53 | 21 | 0.0005 |
| STM01 | 36.0016 | 139.5638 | 1170 | 0 | 0 | | 0 | 2 | 0 | 1.00 | | |
| STM02 | 35.7760 | 139.5827 | 3694 | 15 | 43 | 0 | 0.0116 | 6 | 0 | 1.00 | 0 | 0 |
| STM03 | 35.7973 | 139.6000 | 3468 | 12 | 40 | 0 | 0.0115 | 5 | 0 | 1.00 | 0 | 0 |
| CHB04 | 35.9816 | 139.8335 | 2973 | 50 | 90 | 8 | 0.0303 | 6 | 1 | 0.83 | 6 | 9 | 0.1684 |
| CHB05 | 35.0284 | 139.8457 | 3664 | 25 | 60 | 0 | 0.0164 | 7 | 0 | 1.00 | 0 | 0 |
| CHB07 | 35.4514 | 140.3658 | 2924 | 8 | 22 | 0 | 0.0075 | 5 | 0 | 1.00 | 0 | 0 |
| TKY01 | 35.7682 | 139.8543 | 3545 | 38 | 94 | 4 | 0.0265 | 7 | 1 | 0.86 | 5 | 4 | 0.1715 |
| TKY03 | 35.7106 | 139.6566 | 3477 | 149 | 357 | 39 | 0.1027 | 6 | 4 | 0.33 | 6 | 11 | 0.0014 |
| TKY04 | 35.6106 | 139.7165 | 3467 | 76 | 157 | 16 | 0.0453 | 6 | 1 | 0.83 | 4 | 10 | 0.2427 |
| TKY05 | 35.6269 | 139.6306 | 947 | 0 | 0 | | 0 | 2 | 0 | 1.00 | | |
| TKY06 | 35.6451 | 139.6226 | 3252 | 87 | 158 | 21 | 0.0486 | 5 | 2 | 0.60 | 8 | 13 | 0.0214 |
| TKY07 | 35.6360 | 139.6260 | 1642 | 0 | 0 | | 0 | 3 | 0 | 1.00 | | |
| TKY08 | 35.6059 | 139.6561 | 1446 | 0 | 0 | | 0 | 2 | 0 | 1.00 | | |
| TKY09 | 35.6184 | 139.6329 | 1442 | 0 | 0 | | 0 | 2 | 0 | 1.00 | | |
| TKY10 | 35.6266 | 139.6483 | 1460 | 0 | 0 | | 0 | 2 | 0 | 1.00 | | |
| TKY11 | 35.5999 | 139.6472 | 1459 | 0 | 0 | | 0 | 2 | 0 | 1.00 | | |
| TKY12 | 35.7077 | 139.4513 | 3707 | 27 | 62 | 10 | 0.0167 | 6 | 1 | 0.83 | 10 | 16 | 0.0962 |
| TKY13 | 35.7659 | 139.3633 | 3101 | 8 | 29 | 0 | 0.0094 | 5 | 0 | 1.00 | 0 | 0 |
| TKY15 | 35.6815 | 139.2933 | 953 | 0 | 0 | | 0 | 2 | 0 | 1.00 | | |
| TKY16 | 35.6670 | 139.2365 | 3195 | 4 | 11 | 0 | 0.0034 | 5 | 0 | 1.00 | 0 | 0 |
| KNG01 | 35.5237 | 139.7096 | 854 | 2 | 16 | 0 | 0.0187 | 1 | 0 | 1.00 | 0 | 0 |
| KNG02 | 35.4584 | 139.5284 | 2630 | 13 | 34 | 7 | 0.0129 | 5 | 2 | 0.60 | 31 | 21 | 0.0016 |
| KNG03 | 35.4862 | 139.6183 | 3652 | 77 | 171 | 17 | 0.0468 | 6 | 2 | 0.67 | 7 | 10 | 0.0290 |
| KNG04 | 35.5284 | 139.6681 | 2653 | 56 | 148 | 5 | 0.0558 | 6 | 1 | 0.83 | 3 | 3 | 0.2914 |
| KNG06 | 35.5271 | 139.6317 | 3708 | 17 | 42 | 5 | 0.0113 | 6 | 1 | 0.83 | 15 | 12 | 0.0661 |
| KNG07 | 35.4861 | 139.6298 | 2427 | 31 | 75 | 15 | 0.0309 | 5 | 1 | 0.80 | 6 | 20 | 0.1453 |
| KNG08 | 35.4649 | 139.5790 | 2970 | 16 | 39 | 6 | 0.0131 | 5 | 1 | 0.80 | 15 | 15 | 0.0640 |
| KNG09 | 35.4288 | 139.6473 | 1066 | 0 | 0 | | 0 | 2 | 0 | 1.00 | | |
| KNG11 | 35.2716 | 139.5801 | 3711 | 36 | 76 | 17 | 0.0205 | 7 | 2 | 0.71 | 14 | 22 | 0.1348 |
| KNG12 | 35.2778 | 139.5771 | 921 | 33 | 68 | 0 | 0.0738 | 2 | 0 | 1.00 | 0 | 0 |





| | | | | | | | | | | | | | |
|---|---|---|---|---|---|---|---|---|---|---|---|---|---|
| KNG13 | 35.2956 | 139.5747 | 3540 | 37 | 86 | 6 | 0.0243 | 6 | 1 | 0.83 | 7 | 7 | 0.1372 |
| KNG14 | 35.3308 | 139.5611 | 3702 | 47 | 133 | 24 | 0.0359 | 7 | 3 | 0.57 | 12 | 18 | 0.0015 |
| KNG15 | 35.3211 | 139.5593 | 3092 | 17 | 31 | 0 | 0.0100 | 6 | 0 | 1.00 | 0 | 0 | |
| KNG16 | 35.3062 | 139.5566 | 3397 | 21 | 57 | 6 | 0.0168 | 7 | 1 | 0.86 | 9 | 11 | 0.1117 |
| KNG18 | 35.4146 | 139.4648 | 3708 | 6 | 20 | 0 | 0.0054 | 7 | 0 | 1.00 | 0 | 0 | |
| KNG19 | 35.3328 | 139.4773 | 2452 | 4 | 11 | 0 | 0.0045 | 4 | 0 | 1.00 | 0 | 0 | |
| KNG20 | 35.3144 | 139.4881 | 3580 | 21 | 60 | 6 | 0.0168 | 7 | 1 | 0.86 | 9 | 10 | 0.1116 |
| KNG21 | 35.3345 | 139.4942 | 3610 | 13 | 44 | 0 | 0.0122 | 7 | 0 | 1.00 | 0 | 0 | |
| KNG22 | 35.5991 | 139.2251 | 3469 | 17 | 43 | 23 | 0.0124 | 6 | 2 | 0.67 | 27 | 53 | 0.0022 |
| KNG23 | 35.5933 | 139.2799 | 3609 | 20 | 34 | 17 | 0.0094 | 6 | 1 | 0.83 | 18 | 50 | 0.0552 |
| KNG24 | 35.5575 | 139.3509 | 832 | 11 | 19 | 11 | 0.0228 | 1 | 1 | 0.00 | 44 | 58 | 0.0228 |
| KNG25 | 35.5445 | 139.4341 | 859 | 13 | 20 | 7 | 0.0233 | 1 | 1 | 0.00 | 43 | 35 | 0.0233 |
| KNG26 | 35.5727 | 139.3703 | 2637 | 3 | 10 | 0 | 0.0038 | 4 | 0 | 1.00 | 0 | 0 | |
| KNG29 | 35.4670 | 139.4640 | 3527 | 41 | 83 | 7 | 0.0235 | 6 | 1 | 0.83 | 7 | 8 | 0.1256 |
| KNG31 | 35.3237 | 139.3962 | 3707 | 8 | 15 | 5 | 0.0040 | 7 | 1 | 0.86 | 35 | 33 | 0.0280 |
| KNG32 | 35.3500 | 139.4317 | 3305 | 0 | 0 | | 0 | 6 | 0 | 1.00 | | | |
| KNG33 | 35.4214 | 139.2909 | 3706 | 33 | 60 | 7 | 0.0162 | 7 | 1 | 0.86 | 9 | 12 | 0.1080 |
| KNG34 | 35.4000 | 139.2969 | 1610 | 3 | 10 | 0 | 0.0062 | 3 | 0 | 1.00 | 0 | 0 | |
| KNG35 | 35.4267 | 139.3272 | 3033 | 0 | 0 | | 0 | 6 | 0 | 1.00 | | | |
| KNG36 | 35.4210 | 139.2923 | 2343 | 3 | 10 | 0 | 0.0043 | 4 | 0 | 1.00 | 0 | 0 | |
| KNG37 | 35.3929 | 139.3070 | 1948 | 0 | 0 | | 0 | 3 | 0 | 1.00 | | | |
| KNG38 | 35.4037 | 139.3082 | 2851 | 37 | 66 | 24 | 0.0231 | 6 | 1 | 0.83 | 7 | 36 | 0.1311 |
| KNG39 | 35.3589 | 139.1866 | 1703 | 0 | 0 | | 0 | 2 | 0 | 1.00 | | | |
| KNG40 | 35.3760 | 139.1877 | 1115 | 0 | 0 | | 0 | 2 | 0 | 1.00 | | | |
| KNG46 | 35.3214 | 139.3341 | 3708 | 120 | 225 | 24 | 0.0607 | 6 | 2 | 0.67 | 5 | 11 | 0.0469 |
| KNG47 | 35.3216 | 139.3344 | 2702 | 27 | 42 | 11 | 0.0155 | 5 | 1 | 0.80 | 13 | 26 | 0.0753 |
| KNG48 | 35.3357 | 139.3400 | 3552 | 5 | 14 | 0 | 0.0039 | 7 | 0 | 1.00 | 0 | 0 | |
| KNG49 | 35.3104 | 139.2839 | 2807 | 68 | 140 | 0 | 0.0499 | 4 | 0 | 1.00 | 0 | 0 | |
| KNG50 | 35.2977 | 139.2679 | 2811 | 0 | 0 | | 0 | 4 | 0 | 1.00 | | | |
| KNG51 | 35.3339 | 139.2351 | 1572 | 21 | 49 | 0 | 0.0312 | 3 | 0 | 1.00 | 0 | 0 | |
| KNG53 | 35.3365 | 139.2226 | 2834 | 0 | 0 | | 0 | 4 | 0 | 1.00 | | | |
| KNG54 | 35.3483 | 139.2170 | 2885 | 0 | 0 | | 0 | 4 | 0 | 1.00 | | | |
| KNG55 | 35.3224 | 139.2193 | 951 | 0 | 0 | | 0 | 1 | 0 | 1.00 | | | |





| | | | | | | | | | | | | |
|---|---|---|---|---|---|---|---|---|---|---|---|---|
| KNG57 | 35.3303 | 139.2207 | 2893 | 14 | 28 | 0 | 0.0097 | 4 | 0 | 1.00 | 0 | 0 |
| KNG58 | 35.3224 | 139.1614 | 3707 | 0 | 0 | | 0 | 5 | 0 | 1.00 | | |
| KNG59 | 35.3403 | 139.1591 | 2584 | 93 | 178 | 23 | 0.0689 | 4 | 2 | 0.50 | 7 | 13 | 0.0259 |
| KNG60 | 35.3311 | 139.1550 | 2499 | 34 | 72 | 9 | 0.0288 | 4 | 1 | 0.75 | 9 | 13 | 0.1104 |
| KNG61 | 35.3328 | 139.1169 | 2086 | 0 | 0 | | 0 | 3 | 0 | 1.00 | | |
| KNG63 | 35.3514 | 139.1246 | 3711 | 69 | 131 | 24 | 0.0353 | 4 | 2 | 0.50 | 14 | 18 | 0.0071 |
| KNG66 | 35.4015 | 139.0463 | 835 | 0 | 0 | | 0 | 1 | 0 | 1.00 | | |
| KNG67 | 35.2926 | 139.2345 | 3558 | 4 | 11 | 0 | 0.0031 | 5 | 0 | 1.00 | 0 | 0 |
| KNG68 | 35.2926 | 139.2345 | 3691 | 60 | 140 | 23 | 0.0379 | 5 | 2 | 0.60 | 11 | 16 | 0.0133 |
| KNG69 | 35.2564 | 139.1507 | 2593 | 25 | 41 | 13 | 0.0158 | 3 | 1 | 0.67 | 21 | 32 | 0.0467 |
| KNG70 | 35.2656 | 139.1850 | 1131 | 0 | 0 | | 0 | 1 | 0 | 1.00 | | |
| KNG72 | 35.2461 | 139.1568 | 1520 | 0 | 0 | | 0 | 2 | 0 | 1.00 | | |
| KNG73 | 35.2834 | 139.2242 | 2322 | 19 | 47 | 5 | 0.0202 | 4 | 1 | 0.75 | 12 | 11 | 0.0785 |
| KNG74 | 35.2484 | 139.1406 | 3503 | 0 | 0 | | 0 | 4 | 0 | 1.00 | | |
| KNG75 | 35.2708 | 139.1338 | 2810 | 0 | 0 | | 0 | 2 | 0 | 1.00 | | |
| KNG77 | 35.2711 | 139.0122 | 3256 | 0 | 0 | | 0 | 4 | 0 | 1.00 | | |
| KNG78 | 35.2047 | 139.0307 | 3119 | 0 | 0 | | 0 | 3 | 0 | 1.00 | | |
| KNG80 | 35.2320 | 139.1027 | 3246 | 38 | 53 | 8 | 0.0163 | 3 | 1 | 0.67 | 20 | 15 | 0.0482 |
| SZK01 | 35.2908 | 138.9461 | 2424 | 0 | 0 | | 0 | 3 | 0 | 1.00 | | |
| SZK02 | 35.0963 | 139.0714 | 3452 | 3 | 4 | 4 | 0.0012 | 4 | 1 | 0.75 | 216 | 100 | 0.0046 |
| SZK03 | 35.0460 | 139.0781 | 730 | 17 | 35 | 25 | 0.0479 | 2 | 1 | 0.50 | 10 | 71 | 0.0936 |
| SZK04 | 35.0964 | 139.0720 | 2762 | 11 | 26 | 0 | 0.0094 | 2 | 0 | 1.00 | 0 | 0 |
| SZK05 | 35.0955 | 139.0725 | 3706 | 125 | 249 | 31 | 0.0672 | 4 | 3 | 0.25 | 11 | 12 | 0.0012 |
| SZK06 | 34.9738 | 139.0958 | 3661 | 12 | 38 | 4 | 0.0104 | 4 | 1 | 0.75 | 24 | 11 | 0.0409 |
| SZK07 | 34.9712 | 139.0917 | 2720 | 0 | 0 | | 0 | 3 | 0 | 1.00 | | |
| SZK08 | 35.1450 | 138.7207 | 2686 | 0 | 0 | | 0 | 3 | 0 | 1.00 | | |
| SZK10 | 34.6519 | 138.1691 | 3371 | 10 | 17 | 7 | 0.0050 | 3 | 1 | 0.67 | 66 | 41 | 0.0151 |
| SZK11 | 34.7758 | 138.0078 | 3031 | 0 | 0 | | 0 | 2 | 0 | 1.00 | | |
| SZK13 | 34.7365 | 137.7246 | 1243 | 11 | 32 | 0 | 0.0257 | 2 | 0 | 1.00 | 0 | 0 |
| SZK14 | 34.7363 | 137.7249 | 899 | | | | | | | | | |
| SZK15 | 34.6559 | 137.7735 | 3283 | 11 | 26 | 5 | 0.0079 | 3 | 1 | 0.67 | 42 | 19 | 0.0236 |
| ACH01 | 34.8976 | 137.0043 | 3624 | 3 | 10 | 0 | 0.0028 | 3 | 0 | 1.00 | 0 | 0 |
| ACH02 | 34.9236 | 136.8444 | 3212 | 5 | 12 | 0 | 0.0037 | 3 | 0 | 1.00 | 0 | 0 |





| ACH03 | 35.2492 | 136.9785 | 3673 | 74 | 110 | 45 | 0.0299 | 3 | 1 | 0.67 | 11 | 41 | 0.0898 |
| ACH04 | 35.2586 | 136.9726 | 1432 | 0 | 0 | | 0 | 2 | 0 | 1.00 | | | |
| ACH05 | 35.2702 | 137.0117 | 2969 | 13 | 20 | 0 | 0.0067 | 2 | 0 | 1.00 | 0 | 0 | |
| NGN01 | 36.4780 | 138.1572 | 3180 | 44 | 58 | 33 | 0.0182 | 1 | 1 | 0.00 | 55 | 57 | 0.0182 |
| ISK01 | 36.4118 | 136.4578 | 1715 | | | | | | | | | | |
| EHM01 | 33.4328 | 132.5264 | 819 | 5 | 8 | 8 | 0.0098 | 1 | 1 | 0.00 | 102 | 100 | 0.0098 |
| KMM01 | 32.7936 | 130.7215 | 2730 | 0 | 0 | | 0 | 1 | 0 | 1.00 | | | |

Notes:

$\tau$ is the ratio of alarm days to the total number of observation days. $v$ is the ratio of the number of EQs without preceding anomalies to the

number of predictable EQs. Probability Gain: $PG = (1-v)/\tau$.

### 3.1 Molchan diagram of utilization

Here we discuss how plausible the successful alarm days from the 46 stations were as precursors. Representative verification methods for EQ predictions and forecasts were introduced in Zechar (2010). One method is the Molchan diagram (Molchan, 1991; Molchan and Kagan1992), which plots the fraction of failures to predict EQs "$v$" and the fraction of space-time occupied

by the alarm "$\tau$". The vertical axis $v$ and horizontal axis $\tau$ of the Molchan diagram can be plotted from 0 to 1 using an alarm function given by an alarm threshold. The alarm threshold had been predetermined by the Catfish Club and could not be modified in this study. In addition, the number of predictable EQs at each of the 46 stations was small, ranging from 1 to 7. Therefore, we employed a single set of $v$ and $\tau$ parameters for each station.

The probability of an EQ occurring ($E$) on alarm days ($A$) can be denoted as $P(E|A)$. $P(E|A)$ can be expressed by Bayes theorem

as follows: $P(E|A) = P(A|E) P(E)/P(A)$ (Aki, 1981; Zechar, 2010; Wang et al., 2013). In the equation, $P(A|E)$ is a fraction of the EQs preceded by anomalies over the predictable EQs, $P(E)$ is the unconditional probability of EQ occurrence under each criterion mentioned in the Methods section, and $P(A)$ is the alarm function, i.e., the alarm days ($A$) divided by the number of observed days. From Molchan and Kagan (1992), $v=1-P(A|E)$, and $\tau = P(A)$. Therefore, $P(A|E)/P(A)=(1-v)/\tau$.

Fig. 3 shows $v$ vs $\tau$ for 106 stations because SZK14 and ISK01 had neither anomalous days nor predictable EQs. Thirty-four

stations plotted at $\tau=0$ and $v=1$, which meant that these stations had no alarm days. Twenty-six stations fell above the random guessing line; they had alarm days but no successful alarm days. The other 46 stations under the line denote a probability that exceeded random guessing. The probability gain, $PG=P(A|E)/P(A)$ (Aki, 1981; Molchan and Kagan, 1992; McGuire et al., 2005; Zechar and Jordan, 2008; Wang et al., 2013), of the 46 stations is greater than 1. The maximum $PG$ was 285 at AKT01 and the minimum was 3 at KNG04. Only one station EHM01, had $v=0$, the success rate was 100%. This meant that the station

had no false alarms and no missing cases. In addition, the $PG$ at EHM01 was 102. A feature of the 106 stations was low $\tau$. This finding meant that the experts from the Catfish Club did not readily issue alarms. These low $\tau$ values might contribute to $PG>1$ at 46 stations.





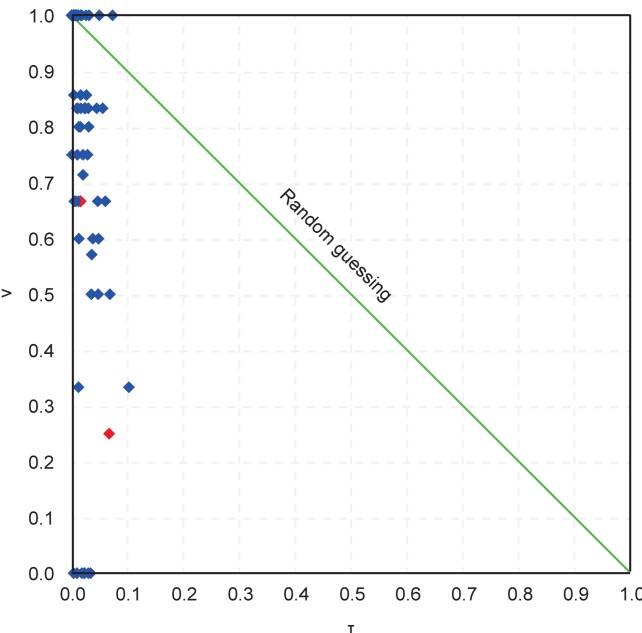

**Fig. 3. Plot of _v_ vs. _τ_ for the 106 stations. The blue diamonds denote the scores of the 104 water level monitoring stations, and the**
**two red diamonds denote the 2 stations where the water temperature was monitored. The green diagonal line represents the random**
**probability threshold. A point above the line suggests that the score is worse than could be obtained by random guessing, while a**
**point below the line indicates that the results are better than random guessing.**

### 3.2 *P-value* evaluation

A *p-value* is the probability of obtaining a result equal to or more extreme than the observed result under the null hypothesis.
In this paper, the *p-value* is obtained from a binomial distribution, i.e., $P(k) = {}_nC_k\, p^k (1-p)^{n-k}$, where $n$ is the number of trials, $k$ is
the number of successful trials, and $p$ is the probability of success. The number of trials $n$ is the number of predictable EQs at
each observation point, the number of successful trials $k$ is the number of EQs with preceding anomalies, and the probability
of success $p$ is the alarm function, i.e., $\tau = P(A)$. Note that the minimum number of trials was one and even the maximum
number of trials was seven. The number of trials may be insufficient.

When the null hypothesis that the "well water anomaly is not an earthquake precursor" was rejected at a value less than 0.05,
the null hypothesis was rejected at 24 of the 46 stations, more than half of the stations. The smallest *p-value* was 0.0005 for
MYG04 with 3 trials.

### 3.3 Evaluation of repeatability

If anomalies appear before multiple EQs in the same area, the likelihood of precursors will be higher. Multiple EQs occurred
in three regions off the coast of Miyagi Prefecture, around Ibaraki Prefecture, and in the Izu region during the observation
period.



1) Off Miyagi Prefecture: Four EQs occurred. The events 2/20/1978M6.7EQ and 6/12/1978M7.4EQ events had two candidate stations, MYG03 and MYG04, and both succeeded in detecting anomalies before the 2 EQs. The 1/19/1981M7.0EQ event had two candidate stations, MYG01 and MYG04 but neither could detect preceding anomalies.

The 12/1/1986M6.0EQ event had one candidate station, MYG01 but the preceding anomalies could not be detected (Fig. 4). The hit rates were 100% for MYG03 (2 out of 2), 66.7% for MYG04 (2 out of 3), and 0% for MYG01 (0 out of 3).

2) Around Ibaraki Prefecture: Two EQs occurred. The 2/27/1983M6.0EQ event had 35 candidate stations. Six of them succeeded in detecting the preceding anomalies. The 10/4/1985M6.0EQ event had 40 candidate stations. None of them detected any preceding anomalies (Fig. 5). No repeatability was observed around the Ibaraki Prefecture area.

3) The Izu region: Three EQs occurred. There were 65 candidate stations for the 1/14/1978M7.0EQ event. Twenty-six of them succeeded in detecting the preceding anomalies. The 6/29/1980M6.7EQ event had 87 candidate stations. Eighteen of them succeeded in detecting the preceding anomalies. There were 35 candidate stations for the 11/22/1986M6.0EQ event. None of them could detect any preceding anomalies. Hence, no station had preceding anomalies for all 3 EQs. Ten stations observed anomalies for both the 1/14/1978M7.0EQ and the 6/29/1980M6.7EQ events. Six of the ten stations

were predictable for the 11/22/1986M6.0EQ event, but the preceding anomalies were not observed (Fig. 6).

Five stations, MYG03, TKY03, KNG02, KNG03, and KNG22 were identified as having preceding anomalies for two predictable EQs. In particular, MYG03 ($PG$=30; $p$-$value$=0.0011) was successful in predicting both of the detectable EQs. In the Izu region, KNG02 ($PG$=31; $p$-$value$=0.0016), KNG03 ($PG$=7; $p$-$value$=0.0290), and KNG22 ($PG$=27; $p$-$value$=0.0022) hit two EQs. On the other hand, the three stations missed all EQs that occurred outside the Izu region even though they were

detectable. This result suggested that even if well water anomalies were precursors to EQs, they were not always observed before EQs with the same focal distance. In other words, each well might only be sensitive to certain epicentral regions.





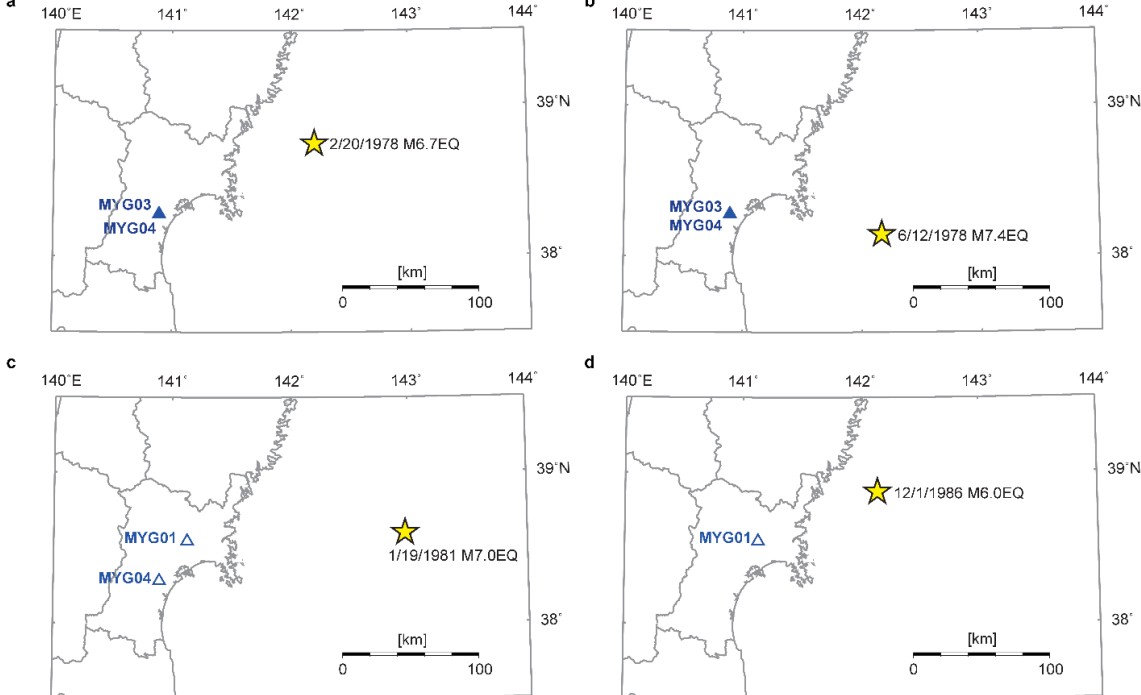

**Fig. 4. Evaluation of the repeatability for the region off the coast of Miyagi Prefecture. The stations where the EQ was detected are indicated by triangle symbols. Solid (open) triangles denote the presence (absence) of a preceding anomaly. Each yellow star shows a predictable EQ: (a) 2/20/1978M6.7EQ, (b) 6/12/1978M7.4EQ, (c) 1/19/1981M7.0EQ, and (d) 12/1/1986M6.0EQ.**


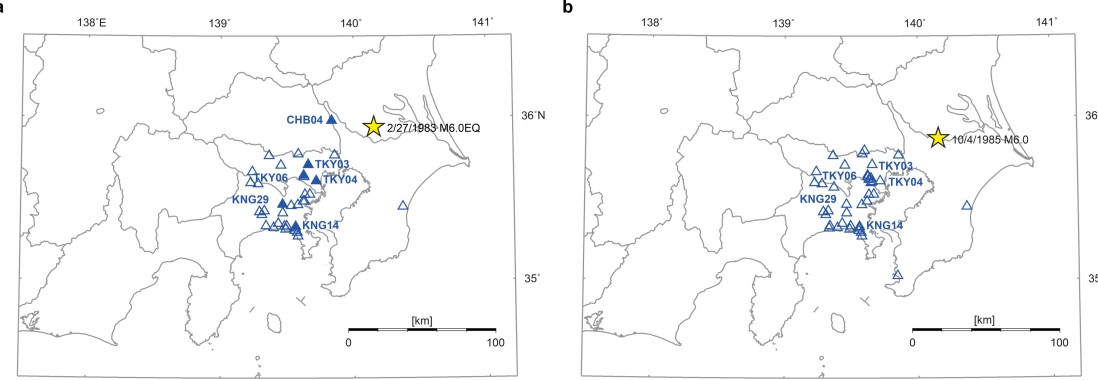

**Fig. 5. Evaluation of the repeatability around Ibaraki Prefecture region. The stations where the EQ was detected are indicated by triangle symbols. Solid (open) triangles denote the presence (absence) of a preceding anomaly. Each yellow star shows a predictable EQ: (a) 2/27/1983M6.0EQ and (b) 10/4/1985M6.0EQ.**



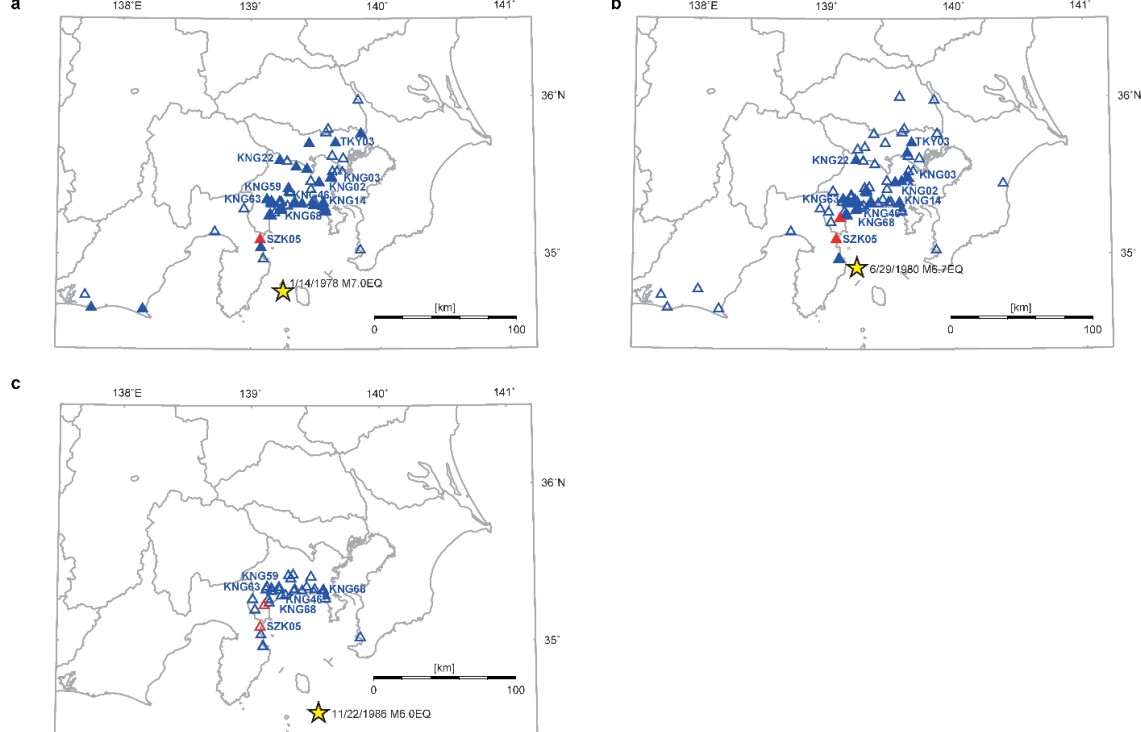


**Fig. 6. Evaluation of the repeatability in the Izu region. The stations where the EQ was detected are indicated by triangle symbols. Solid (open) triangles denote the presence (absence) of a preceding anomaly. The blue and red triangles show the water level and water temperature observation stations, respectively. Each yellow star shows a predictable EQ. (a) 1/14/1978M7.0EQ, (b) 6/29/1980M6.7EQ, and (c) 11/22/1986M6.0EQ.**

### 3.4 How did well water anomalies appear?

Wakita (1978) gathered the groundwater anomalies of 113 EQs from 684 to 1978 and suggested that it was difficult to identify the general regularity of how water level, water temperature, and muddiness anomalies appeared. In this study, some wells exhibited anomalies, whereas others did not, even if the wells were very close. For instance, KNG14 recorded three EQ precursors, while KNG15, located nearby, recorded none.

Umeda et al. (2010) described a well water depression prior to the 1946 Nankai EQ (*M*8.0). The wells were in a small delta along the Pacific coast. These authors suggested that the well locations caused the groundwater to drop. On the other hand, the 46 stations that had successful alarm days were located at various sites. The anomalous changes noted in this study could not be explained by Umeda et al. (2010). The mechanisms behind these well water anomalies remain unclear.

### 3.5 Is the folklore true?

Water level change values were associated with most anomaly days during the first half of the observation period recorded in the newsletters. We conducted a thought experiment to assess whether the well users could determine if a fluctuation was



unusual, using these values. We assumed that they comprehended the groundwater level qualities as well as the experts of the Catfish Club and presumed that the anomaly threshold was 30 centimetres (cm). The following five EQs had associated values before occurrences:

1) 1/14/1978M7.0EQ (Izu Ohshima): Twenty-four of the stations with successful hits recorded fluctuating values, and only KNG63 (*PG*=14; *p-value*=0.0071) exceeded the threshold.

    2) 2/20/1978M6.7EQ (Off Miyagi): MYG03 and MYG04 had fluctuating values, and MYG04 (*PG*=53; *p-value*=0.0005) exceeded the threshold.

    3) 6/12/1978M7.4EQ (Off Miyagi): MYG03 and MYG04 had fluctuating values, and MYG03 (*PG*=30; *p-value*=0.0011)
exceeded the threshold.

    4) 7/13/1979M6.0EQ (Yamaguchi Prefecture): EHM01 was the only candidate station, however, the anomaly value of at +20 cm was below the threshold.

    5) 6/29/1980M6.7EQ (Izu Peninsula): There were fifteen stations with successful hits and fluctuating values, and 6 stations, TKY06, KNG22, KNG38, KNG59, KNG63, and KNG69, exceeded the threshold.

This attempt suggests that the precursors of 4 out of 5 EQs could be regarded as macroscopic anomalies based on well water level changes. Only KNG63 (*PG*=14; *p-value*=0.0071) exceeded the threshold before multiple EQs. In addition, we investigated the probability of a false alarm, which meant that fluctuations were over the threshold despite no EQs occurring. Table 3 shows the maximum values of the 68 stations that had associated values. False alarms, defined as anomalies not associated with EQs, were confirmed at 33 of 39 stations. These 33 stations were considered to have issued at least one false
alarm during the observation period. In addition, even KNG63, which had detected precursors before two EQs, had three false alarms. These results suggested that there were many false alarms unrecorded in the old documents and in the folklore.

Regarding missing cases (i.e., EQs without anomalies), a total of 195 potentially predictable EQs occurred at the 39 stations, of which 156 were not preceded by any anomalies. Therefore, the overall missing rate was 80%. This implies that there were also many missing cases behind successful accounts.

The experiment suggested that well water anomalies recorded prior to impending EQs in the old documents and folklore might include "true" precursors. However, unrecorded false alarms and missing cases might have frequently occurred. Some critics have claimed that successful cases were only by chance, and the same phenomena might occur without EQs. The study, based on the observation data, indicated that the claim of the former was false, while the claim of the latter was true.

250                        **Table 3. The maximum values of anomalies from 68 stations.**

| Station name | Maximum value [cm] | Station name | Maximum value [cm] |
|---|---|---|---|
| AKT01 | -10 | KNG23 | -10 |
| MYG01 | 11 | KNG24 | 28 |
| MYG03* | ***31*** | KNG25* | 18 |



| | | | | |
|---|---|---|---|---|
| MYG04* | *-110* | | KNG29 | *56* |
| STM02 | *120* | | KNG31* | -18 |
| STM03 | *70* | | KNG33 | *82* |
| CHB04 | *235* | | KNG34 | *75* |
| CHB05 | *-48* | | KNG36 | *30* |
| CHB07 | 11 | | KNG38 | *58* |
| TKY01 | *56* | | KNG46 | 24 |
| TKY03 | *90* | | KNG47 | 14 |
| TKY04 | *135* | | KNG48 | *38* |
| TKY06 | *188* | | KNG49 | 20 |
| TKY12 | 10 | | KNG51 | 16 |
| TKY13 | *160* | | KNG57 | 18 |
| TKY16 | *60* | | KNG59* | *117* |
| KNG01 | 18 | | KNG60 | *40* |
| KNG02* | 22 | | KNG63* | *122* |
| KNG03 | *166* | | KNG67 | -10 |
| KNG04 | *185* | | KNG68 | *72* |
| KNG06 | *70* | | KNG69 | *60* |
| KNG07 | 20 | | KNG73 | *65* |
| KNG08* | 17 | | SZK02* | *70* |
| KNG11 | *40* | | SZK04 | *65* |
| KNG12 | 23 | | SZK06 | *43* |
| KNG13 | 28 | | SZK10 | 7 |
| KNG14 | *80* | | SZK13 | *-58* |
| KNG15 | *130* | | SZK15 | 28 |
| KNG16* | 14 | | ACH01 | -10 |
| KNG18 | *36* | | ACH02 | 15 |
| KNG19 | 5 | | ACH03 | *35* |
| KNG20* | -26 | | ACH05 | *35* |
| KNG21 | *60* | | NGN01 | 25 |
| KNG22* | *-32* | | EHM01* | 20 |

Notes:



The asterisk * attached to the "Station name" for the 13 stations denotes cases where the maximum value appeared prior to an EQ occurrence. The 39 stations numbered in bold italics were those where the threshold value was exceeded. False alarms, anomalies without associated EQs, were confirmed at 33 of 39 stations.

## 3.6 Feasibility of earthquake prediction methods

Eleven years of observation data from multiple stations suggested the plausibility of using well water anomalies as EQ precursors. Indeed, EHM01 (*PG*=102; *p-value*=0.0098) showed no false alarms and no missing cases (1 of 1) and MYG03 (*PG*=30; *p-value*=0.0011) indicated that *P(A/E)* equalled 1 for multiple EQs (2 of 2). However, only one successful and 819 observation days at EHM01 might be too few and too short a period, and two successful observations at MYG03 might also be too few to empirically confirm whether the anomalies were "true" precursors. The findings of this study imply that if one continues observations for a few decades, one may recognize a credible well. Then, if one can maintain observations for hundreds of years, one may predict the next destructive EQ; however, it is difficult to sustain consistent monitoring. The reports of the "Black and White" Score Table by the Catfish Club had already been completed. These are the reasons why an empirical EQ prediction method has not ever been developed. Therefore, empirical research on EQ precursors for disaster mitigation requires ongoing support.

## 4 Conclusions

The present study revealed that some well water anomalies were statistically significant EQ precursors, even in shallow wells (i.e., unconfined groundwater). Hence, you may identify a precursor prior to the next EQ. Conversely, the data suggested that many false alarms and missing cases also occurred. These results imply that the macroscopic anomalies of well water recorded in old documents and folklore contain elements of truth, while many unrecorded false alarms and missed events are obscured by successful accounts. We also confirmed that eleven years of observation was too short to clarify whether an anomaly was a real precursor.

*Acknowledgements.* The authors would like to thank the Catfish Club, Hot Springs Research Institute of Kanagawa Prefecture, Japan for their efforts to observe the data, and the anonymous employees for their help in digitalizing the data. The maps in this study were generated using the Generic Mapping Tools (GMT; Wessel et al., 2019). This study was funded by ERI JURP 2020-KOBO08, ERI JURP 2023-KOBO30, and ERI JURP 2024-KOBO03 at the Earthquake Research Institute, the University of Tokyo.



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

**Code availability**

We did not use original code to transcribe the paper-based data into digital format. We typed the code in using Excel by
Microsoft.

**Data availability**

The original data are included in the newsletters of the Catfish Club by the Hot Springs Research Institute of Kanagawa
Prefecture, Japan. In addition, the data protect the privacy of the member of the Catfish Club. Therefore, the author cannot
share the data. However, you can see the data in the newsletters if you visit to the Kanagawa Prefectural Library in Japan.

**Author contributions**

Y.O. designed the study, analysed the data, and wrote the manuscript.

**Competing interest**

The author declares that there are no competing interests.

Correspondence and requests for materials should be addressed to
orihara@u-gakugei.ac.jp