# Peer review of "Statistical Evaluation of Well Water Anomalies as Potential Precursors to Large Earthquakes"

_EGUsphere, 2025_

## Referee Comment (RC2)

**General characteristics of the manuscript**

The manuscript is devoted to the analysis of materials from daily observations at 108 near-surface wells (depths up to 5-20 m), which were carried out by volunteers over 11 years from February 1, 1977 to March 31, 1987. The source materials are tables containing daily assessments of the presence or absence of anomalies in water level changes in the form of black or white symbols, respectively. This volunteer group was called the "Catfish Club" and was organized by the Kanagawa Prefectural Hot Spring Research Institute, Japan.

The author selected 18 earthquakes (Table 1) based on a specific criterion (p. 4) that takes into account the relationship between earthquake magnitude and its epicentral distance to the stations. It was assumed that these earthquakes could be accompanied by precursor anomalies in water level changes.

The alarm time was determined from the date of the anomaly in water level changes plus 7 days. When anomalies were observed consecutively, all anomaly days and the continuous 7 days were considered alarm days. If another anomaly appeared before the end of the 7-day interval, the alarm days continued. The diagram for dividing time periods into "false alarm days" and "successful alarm days" is shown in Fig. 1.

Further statistical processing of observation materials at individual stations for the purpose of assessing the effectiveness of the manifestation of anomalous changes in water levels (black symbols) is traditional and is presented in the section 3 and in Table 2.

As a result, the author argues that some well water anomalies were statistically significant precursors of earthquakes. However, numerous "missed targets" occurred.

The overall conclusion of the study is quite ambiguous and boils down to the fact that macroscopic well water anomalies (rises and falls in water level with amplitudes of tens of centimeters) contain elements of truth, along with a large number of recorded "false alarms" and "missed earthquakes (targets)."

**General impression of this work**

I was interested to learn about the history of volunteer work in Japan on monitoring near-surface groundwater to detect anomalies preceding strong earthquakes. I also found the presentation of a methodology for statistically processing formalized data on the presence/absence of "anomalies—black symbols"—in a well network in relation to earthquakes (Table 1) helpful.

At the same time, there are a number of comments regarding the lack of description of the observation wells, methods of processing primary information by experts to decide on the presence of an anomaly (black symbol) and many other details that were later clarified during specialized observations of changes in the level of groundwater.

(1) It is currently generally accepted that the effects of individual earthquakes do not manifest themselves in changes in the water level of near-surface groundwater. The exception is co- and postseismic effects resulting from the liquefaction of loose sedimentary deposits during noticeable shaking during strong earthquakes.

However, the manuscript lacks data on the composition of water-bearing rocks in individual wells, as well as on the intensity of shaking during individual earthquakes.

It follows from the text and the bibliography that the author is unfamiliar with the results of precision well water level observations (observation frequency  $\geq 1$  Hz), which are widely conducted in Japan, the USA, China, Russia, and other countries since the end of the twentieth century.

These high-frequency observations have shown that the effects of strong earthquakes can include not only precursors but also changes in groundwater level/pressure due to the dynamic

action of seismic waves, as well as coseismic effects due to changes in the static stress state of water-bearing rocks. When a well is located in the near zone of a strong earthquake, co-seismic effects in changes in water levels can be significant and long-lasting. Therefore, the time of observation of the effect of changes in the water level at the well with an accuracy of minutes-seconds is of great importance as well as indicating the time of earthquakes.

These aspects of water level/pressure fluctuations during strong earthquakes are not reflected in the manuscript under review. However, a critical approach to the source data used for statistical analysis is necessary in this work. The study doesn't pay enough attention to linking the timing of earthquakes and recorded anomalies. Therefore, the nature of the anomalies, even if they occurred, is unknown.

Unfortunately, this article promotes the false and outdated notion that all anomalies in well water level changes are precursors to earthquakes. Therefore, this manuscript in its present form is not recommended for publication.

However, with some revision, this material may be of interest to the scientific community interested in the historical aspect of studying earthquake precursors and the need for statistical processing of data on them.

**Recommendations**

1. The title of the work «Statistical Evaluation of Well Water Anomalies as Potential Precursors to Large Earthquakes» should be changed to remove the mention of precursors.

The term "precursors" can be replaced by "identified anomalies" or another identical one, referring to the black symbols in the Score Table, which is more consistent with reality, taking into account modern ideas about the impact of seismicity on groundwater and significant shortcomings in the observation methodology in 1977-1987.

- 2. The Introduction should provide an overview of current understanding of seismicity effects in groundwater based on well observations in Japan and around the world. This will allow for a critical assessment of the work of volunteers and experts between 1977 and 1987.
- 3. It would be desirable to provide an additional table with more detailed characteristics of individual points and observation times at each station. Table 2 provides only their coordinates.
- 4. It is advisable to compare the earthquake data from the JMA in Table 1 with data from international catalogs. Specifically, the number of earthquakes with Mw = 6.0 or greater for the region under consideration from the NEIS GSUS catalog was 30 events (?).
- 5. It is necessary to supplement the list of references with modern works on the problem under consideration.

After eliminating the aforementioned uncertainties regarding the stations, the method for identifying "anomalies," and clarifying the relationship with earthquake timing and intensity, the results of the statistical analysis could have scientific significance.